# FCSwinU: Fourier Convolutions and Swin Transformer UNet for Hyperspectral and Multispectral Image Fusion

**DOI:** 10.3390/s24217023

**Published:** 2024-10-31

**Authors:** Rumei Li, Liyan Zhang, Zun Wang, Xiaojuan Li

**Affiliations:** 1College of Resource Environment and Tourism, Capital Normal University, No. 105, North Road of West 3rd Ring, Beijing 100048, China; 2220902206@cnu.edu.cn (R.L.); 2230902185@cnu.edu.cn (Z.W.); lixiaojuan@cnu.edu.cn (X.L.); 2Key Laboratory of 3-Dimensional Information Acquisition and Application, Ministry of Education, Capital Normal University, No. 105, North Road of West 3rd Ring, Beijing 100048, China

**Keywords:** hyperspectral image (HSI), multispectral image (MSI), image fusion, deep learning, Swin Transformer

## Abstract

The fusion of low-resolution hyperspectral images (LR-HSI) with high-resolution multispectral images (HR-MSI) provides a cost-effective approach to obtaining high-resolution hyperspectral images (HR-HSI). Existing methods primarily based on convolutional neural networks (CNNs) struggle to capture global features and do not adequately address the significant scale and spectral resolution differences between LR-HSI and HR-MSI. To tackle these challenges, our novel FCSwinU network leverages the spectral fast Fourier convolution (SFFC) module for spectral feature extraction and utilizes the Swin Transformer’s self-attention mechanism for multi-scale global feature fusion. FCSwinU employs a UNet-like encoder–decoder framework to effectively merge spatiospectral features. The encoder integrates the Swin Transformer feature abstraction module (SwinTFAM) to encode pixel correlations and perform multi-scale transformations, facilitating the adaptive fusion of hyperspectral and multispectral data. The decoder then employs the Swin Transformer feature reconstruction module (SwinTFRM) to reconstruct the fused features, restoring the original image dimensions and ensuring the precise recovery of spatial and spectral details. Experimental results from three benchmark datasets and a real-world dataset robustly validate the superior performance of our method in both visual representation and quantitative assessment compared to existing fusion methods.

## 1. Introduction

Due to the limited spectral resolution of panchromatic (PAN) or multispectral imaging (MSI), material identification poses significant challenges across various fields [1,2]. For instance, distinguishing between true and false targets in military operations remains difficult [3], and precision agriculture struggles to assess soil fertility or crop water stress with high accuracy [4]. Similarly, in medical diagnostics, multispectral imaging often lacks the precision to detect subtle lesions [5].

Hyperspectral imaging (HSI) captures detailed spectral information across hundreds of contiguous bands, enabling precise material identification and offering more insights than PAN or MSI [6]. This makes HSI especially useful for improving the accuracy of quantitative remote sensing by distinguishing materials based on their spectral signatures. However, current hyperspectral imaging technology struggles to balance spatial and spectral resolution. High spectral resolution often reduces spatial clarity, limiting its effectiveness in applications requiring both [7]. Fortunately, modern sensors now produce MSI with high spatial resolution, which opens up new possibilities for overcoming this limitation. The fusion of hyperspectral and multispectral images has emerged as a promising technique, combining the spectral richness of HSI with the spatial detail of MSI to improve overall image quality and expand the practical applications of hyperspectral data [8,9].

Traditional approaches leverage prior knowledge like self-similarity, sparsity, and low-rank constraints [10,11]. However, these handcrafted priors are dataset-specific and often require adjustment, limiting their generalization and fusion performance. With the rise of computer vision [12], deep learning methods, particularly CNNs, have outperformed traditional fusion techniques [13,14]. Although CNNs are highly effective in extracting spatial information, they struggle to model global semantic interactions and contextual information due to the inherent limitations of convolutional operations. This shortcoming is particularly problematic in the fusion of LR-HSI and HR-MSI, where the scale differences between the images introduce ill-posed challenges [15,16]. The Transformer [17] exhibits powerful capabilities in capturing long-range dependencies and is suitable for extracting globally fused features from images. Recently, inspired by the success of Vision Transformer (ViT) [18] in the field of computer vision, researchers have attempted to introduce ViT into the domain of hyperspectral image fusion [19], achieving competitive fusion results. However, the traditional ViT divides the image into fixed-size patches and employs these patches as tokens for self-attention computation, aiming to reduce computational complexity. This approach limits their ability to represent spatial features at different scales and restricts their capability to extract spectral features from hyperspectral images. Nevertheless, ViT lacks certain important attributes present in traditional convolutional neural networks, such as locality and translational invariance [20]. In contrast, Swin Transformer [21] demonstrates advantages in handling cross-scale features and requires less computation compared to ViT. It effectively processes images with rich spatial details and extracts spectral features from hyperspectral images. Consequently, models based on the Swin Transformer have made significant progress in the field of natural image super-resolution [22] and have gradually been introduced into the domain of hyperspectral and multispectral fusion. The potential of the Swin Transformer in the field of hyperspectral and multispectral fusion is still in its early stages.

Many deep learning-based techniques first interpolate LR-HSI to match the HR-MSI scale and then apply CNN for local feature extraction. However, this interpolation step can introduce noise, impacting image quality [23]. Inspired by the application of FFC in various studies [24,25,26], we propose an SFFC structure that pre-extracts global features from LR-HSI to overcome these challenges. While existing fusion methods with the Swin Transformer often focus on single-scale feature extraction, they overlook the crucial cross-scale information interaction necessary for fusion tasks. We introduce a multi-scale feature reminiscent of the UNet encoder–decoder structure, facilitating seamless information exchange between the encoder and decoder, thereby enhancing fusion quality across various scales. The main contributions of this article can be summarized in three key points.

To fully adapt to the significant scale difference between LR-HSI and HR-HSI and overcome the limitations of local feature learning, we propose a novel hyperspectral and multispectral fusion network called FCSwinU. It utilizes a windowed self-attention mechanism to extract global semantic fusion features and introduces a multi-scale feature through an encoder–decoder structure. This architecture allows information to be propagated and fused across different scales through skip connections between the encoder and decoder.We introduce the SFFC block specifically designed for utilizing global information in fusion tasks. SFFC is based on FFC and is used for pre-extracting feature representations of LR-HSI. It enables the extraction of more comprehensive, detailed, and stable features. SFFC consists of two branches: a spatial branch and a frequency branch. We leverage FFC to extract global information in the frequency branch and residual modules in the spatial branch to enhance local feature representation, compensating for the loss of LR-HSI interpolation information.We propose SwinTFAM to achieve adaptive extraction of hyperspectral and multispectral information. We construct the SwinTFRM module in the decoder part for feature recombination and use a final feature reconstruction module to achieve high-quality fusion and restore the original image size while preserving important details.

## 2. Related Work

Recent advancements in deep learning have significantly impacted image processing, particularly in the fusion of LR-HSI and HR-MSI. One of the pioneering works in this domain is by Dian et al. [27], who proposed a novel approach that learns image priors and incorporates them into the fusion framework. They achieve this by solving the Sylvester equation to initialize HR-HSI, surpassing traditional fusion methods [28,29]. Building upon this, Zhou et al. [30] introduced the pyramid fully convolutional network (PFCN), which employs two sub-networks to extract and fuse features from LR-HSI and HR-MSI. To mitigate the issue of overlooking shallow features in HR-MSI, Xiao et al. [31] proposed the Dual-UNet fusion method, utilizing an encoder–decoder network to extract multi-scale HR-MSI features, which are then injected into a UNet structure. Additionally, the attention multi-hop graph and multi-scale convolution fusion network (AMGCFN) [32] addresses the flexibility required for irregular patterns and pixel-level feature loss. Several other methods, such as the multi-scale difference and variation feature fusion network (MSDFFN) [33,34], have also been developed, incorporating depth attention networks for effective hyperspectral and multispectral image fusion. However, many CNN-based methods primarily focus on local feature extraction, often neglecting global features essential for single-scale fusion problems. This oversight can lead to fusion results that lack comprehensive global contextual information, ultimately limiting their expressiveness and accuracy.

In contrast, the Transformer architecture, initially proposed for machine translation tasks [17], employs a self-attention mechanism that models global dependencies, facilitating deep information propagation and allowing for powerful global feature extraction. Its applications have extended into natural language processing [35,36], and more recently, into computer vision. Recent advancements in Transformer-based models specifically targeting image fusion include Wang et al.’s multi-layer Cross-Transformer (MCT-Net) [37], which integrates a cross-attention mechanism to address long-range spatial dependencies during the fusion process, effectively combining spatial and spectral information. Similarly, Chen et al. [38] proposed a multi-scale Transformer that merges the advantages of CNNs for local spatial–spectral processing with the global capabilities of Transformers. Li et al. [39] introduced cross-scale and SW-CSNA modules to capture global and patch-level similarities, enhancing the learning of both local and non-local features.

Despite these innovations, ViTs have limitations in local feature retention due to their requirement to divide input images into fixed-size patches, leading to potential information loss. To address this, Jia et al. [40] introduced multi-scale features and proposed a novel spatial–spectral multi-scale Transformer network, enhancing the richness of feature extraction. Interactformer [41] utilizes the Swin Transformer to better interact with both global and local features, although it still encounters challenges in single-scale feature extraction. Additionally, Ma et al. [42] leveraged the window attention mechanism of the Swin Transformer as a deep prior network, successfully learning implicit prior information from hyperspectral images for improved fusion outcomes.

In response to the challenges posed by scale differences in LR-HSI and HR-HSI fusion, our research introduces a novel approach by combining self-attention mechanisms across different scales with windowed self-attention techniques. This method effectively captures long-range dependencies and local structures within images while preserving global contextual information [43]. Our FCSwinU method stands out as the first to integrate Swin Transformer blocks within a multi-scale architecture specifically for hyperspectral and multispectral image fusion. This approach addresses the limitations of single-scale extraction and significantly enhances fusion quality.

## 3. Methodology

### 3.1. Model Formulation

#### 3.1.1. Mathematical Representation of Hyperspectral Images

Hyperspectral images typically present a three-dimensional cube structure and are represented as a three-dimensional tensor with the following dimensions: height × width × number of spectral bands. Each spectral band contains specific spatial information, displaying data of the same scene across different spectral intervals. In this paper, lowercase letters represent scalars, bold letters represent matrices, and calligraphic letters are used to denote tensors. We denote $Y\in R^{H\times W\times s}$ as the HR-MSI, and $Z\in R^{h\times w\times S}$ as the LR-HSI. The goal of hyperspectral and multispectral fusion is to effectively combine information from both LR-HSI, which has high spectral resolution with *S* spectral bands but lower spatial resolution, and HR-MSI, which has high spatial resolution with width and height, *H* and *W*, respectively, but lower spectral resolution, *s* (h<H, w<W, s<S). Our objective is to fuse HR-MSI and LR-HSI to produce an HR-HSI $\hat{X}\in R^{H\times W\times S}$ that possesses both high spatial and spectral resolutions.

#### 3.1.2. Fast Fourier Transform Convolution

Fast Fourier transform convolution [24] is a technique for efficiently implementing convolution computations using fast Fourier transform [44] (FFT). In the FFT convolution process, the input signal and the convolution kernel are first transformed into frequency domain representation using the discrete Fourier transform (DFT). The DFT representations are as follows, where X[k] represents the input signal and H[k] the convolution kernel:(1)$$X\left[k\right]=\sum_{n=0}^{N-1}x\left[n\right]e^{-j2\pi kn/N}$$
(2)$$H\left[k\right]=\sum_{n=0}^{N-1}h\left[n\right]e^{-j2\pi kn/N}$$
where x[n] and h[n] are the discrete samples of the input signal and the convolution kernel, respectively, and their frequency domain representations are given by Equations (Equation 1) and (Equation 2). *N* denotes the signal length, and *j* is the imaginary unit. In the frequency domain, FFT convolution computes the frequency domain representation of the convolution result Y[k] by point-wise multiplication:(3)Y[k]=X[k]·H[k]

Finally, the inverse discrete Fourier transform (IDFT) is applied to the product result Y[k] to convert it back into the time domain, yielding the final convolution result y[n]:(4)$$y\left[n\right]=\frac{1}{N}\sum_{k=0}^{N-1}Y\left[k\right]e^{j2\pi kn/N}$$

The main advantage of FFT convolution, as described by Equation (Equation 3), is that it enables efficient multiplication operations using fast Fourier transform algorithms such as FFT. Compared to direct convolution in the time domain, FFT convolution reduces the computational complexity from $O(N^{2})$ to O(NlogN) by performing multiplication operations in the frequency domain, where *N* is the signal length.

### 3.2. Proposed FCSwinU

In this study, an end-to-end hyperspectral and multispectral fusion network, termed FCSwinU, is proposed, with its structure illustrated in Figure 1. The architecture of the network primarily comprises a spectral fast Fourier convolution (SFFC) module, a deep feature extraction module, and an image reconstruction module. The SFFC module mainly consists of convolutional layers and the fast Fourier transform (FFT). The deep feature extraction module integrates the Swin Transformer network with a UNet-like architecture [45], encompassing the Swin Transformer feature abstraction module (SwinTFAM) in the encoder and the Swin Transformer feature reconstruction module (SwinTFRM) in the decoder. The image reconstruction module is formed by two convolutional layers and the LeakyReLU activation function between them.

Firstly, considering the trade-off between performance and processing speed, we employ a bilinear interpolation method to upsample the LR-HSI, obtaining a high-resolution hyperspectral image, $Z_{up}\in R^{H\times W\times S}$, as represented by Equation (Equation 5):(5)$$Z_{up}=Up\left(Z\right)$$
Up(·) denotes the bilinear interpolation upsampling function. Subsequently, we feed $Z_{up}$ into the SFFC module to pre-extract features from the low-resolution hyperspectral image, leading to the generation of $Z_{out}$.

Next, as shown in Figure 1, we concatenate the obtained $Z_{out}$ with the HR-MSI to form a merged feature map $D\in R^{H\times W\times (S+s)}$, described by Equation (Equation 6):(6)$$D=Concat(Z_{out},Y)$$
In Equation (Equation 6), Concat(·) represents concatenation along the channel dimension. The merged feature map *D* is then input to the deep feature extraction module for further extraction and fusion of hyperspectral and multispectral information.

As depicted in Figure 1, for the encoder part, we divide the input feature map *D* into non-overlapping 4 × 4 blocks to transform it into sequence embeddings. This block-wise approach results in a feature dimension of 4×4×(S+s)=16(S+s) for each block. Additionally, a linear embedding layer is employed to project the feature dimension into an arbitrary dimension *C*. The transformed block tokens pass through multiple SwinTFAM and feature abstraction (FA) layers to generate hierarchical feature representations. Specifically, the FA layer handles downsampling and dimension augmentation, while the Swin Transformer is responsible for learning feature representations.

The decoder consists of SwinTFRM and feature reconstruction (FR) layers. Through skip connections, multi-scale features obtained from the encoder are fused with contextual features to compensate for spatial information loss due to downsampling. In contrast to the FA layer, the FR layer is specifically designed for upsampling. It reshapes feature maps of adjacent dimensions into larger feature maps with twice the upsampling resolution. The final FR layer is used for fourfold upsampling.
(7)F=Upsample_4x(H)
$F\in R^{H\times W\times C}$ represents the deep spatial–spectral features, where the upsample_4x operation is used for performing a 4x upsampling. $H\in R^{\frac{H}{4}\times \frac{W}{4}\times C}$ indicates the feature map representation in the previous layer of the network. We restore these feature maps to the original input resolution of H×W, thus obtaining the deep spatial–spectral features *F*.

In the image reconstruction module, we first adjust the depth of the input feature map *F* to S+s through a convolution operation, followed by a nonlinear transformation using the LeakyReLU activation function. Subsequently, another convolution layer adjusts the depth of the feature map to *S*, resulting in the estimated image outcome $\hat{X}\in R^{H\times W\times S}$.
(8)$$\hat{X}=IR\left(F\right)$$
IR(·) denotes the image reconstruction function, implementing the main functionality of the image reconstruction module. Finally, we optimize the network parameters by minimizing the $l_{1}$ loss function.
(9)$$l_{1}=∥\hat{X}-X∥$$
where $X\in R^{H\times W\times S}$ is the reference image of the true high-resolution hyperspectral imagery.

### 3.3. Spectral Fast Fourier Convolution

In the SFFC network architecture, as depicted in Figure 2, the left branch is a traditional spatial convolution module, and the right branch is a fast Fourier convolution module. The spatial convolution module on the left utilizes a classic residual module structure, enhancing the expressiveness of the model by incorporating residual connections and convolution layers. Specifically, the module is described as follows:
(10)$$Z_{out}=\mathrm{Conv}\left(\mathrm{concat}\left(C_{spa}\left(Z_{up}\right),C_{spe}\left(Z_{up}\right)\right)\right)$$
(11)$$C_{spa}\left(Z_{up}\right)=H_{crc}\left(Z_{up}\right)+Z_{up}$$
(12)$$X=H_{cbr}\left(Z_{up}\right)$$
(13)$$C_{spe}\left(Z_{up}\right)=H_{c}\left(H_{fft}\left(X\right)+X\right)$$

The feature map of the upsampled low-resolution hyperspectral image, denoted as $Z_{up}\in R^{H\times W\times S}$, is fed into two distinct processing branches: spatial and frequency domains. The spatial convolution module in the spatial domain is represented as $C_{\mathrm{spa}}\left(\cdot \right)$, as detailed in Equation (Equation 11), while the fast Fourier convolution module in the frequency domain is denoted as $C_{\mathrm{spe}}\left(\cdot \right)$, refer to Equation (Equation 13). $H_{\mathrm{crc}}\left(\cdot \right)$ represents the head and tail 3×3 convolution layers, involving LeakyReLU operations; $H_{\mathrm{cbr}}\left(\cdot \right)$ includes convolution layers, BatchNorm (BN) layers, and ReLU operations; $H_{\mathrm{fft}}\left(\cdot \right)$ comprises a series of channel-based 2D FFT, frequency convolution, and ReLU operations, followed by inverse 2D FFT. As illustrated in Figure 2, the process starts with a real-part Fourier transformation of the feature map *X*, converting it into a complex form by stacking the real and imaginary parts separately, thus transforming the traditional spatial–spectral features into the frequency domain for global feature extraction. Subsequently, the spectral dimension is reduced through convolution operations, followed by nonlinear activation and tensor dimension swapping. Finally, the real and imaginary parts are merged back into a complex tensor, and the original size of *X* is restored through inverse Fourier transformation. The concat(·) function combines the results of spatial convolution and fast Fourier convolution, resulting in $Z_{\mathrm{out}}\in R^{H\times W\times 2S}$. Ultimately, a 1×1 convolution operation is employed for projection transformation, restoring the spectral dimension to the original hyperspectral dimension, yielding the final output of the SFFC module, $Z_{\mathrm{out}}\in R^{H\times W\times S}$.

### 3.4. Swin Transformer Feature Abstraction and Reconstruction Module

Unlike the traditional multi-head self-attention (MSA) module, the Swin Transformer block is constructed based on a sliding window mechanism. As shown in Figure 3a, two consecutive Swin Transformer blocks are presented. Each block consists of a LayerNorm (LN) layer, a multi-head self-attention module, a residual connection, and a two-layer MLP with GELU non-linear activation function. The windowed multi-head self-attention (W-MSA) module and the shifted window multi-head self-attention (SW-MSA) module are applied in these two Transformer blocks, respectively. Based on the window partitioning mechanism, the consecutive Swin Transformer blocks can be represented as follows:(14)$${\hat{g}}_{l}=W-MSA\left(\mathrm{LN}\left(g_{l-1}\right)\right)+g_{l-1}$$
(15)$$g_{l}=MLP\left(\mathrm{LN}\left({\hat{g}}_{l}\right)\right)+{\hat{g}}_{l}$$
(16)$${\hat{g}}_{l+1}=SW-MSA\left(\mathrm{LN}\left(g_{l}\right)\right)+g_{l}$$
(17)$$g_{l+1}=MLP\left(\mathrm{LN}\left({\hat{g}}_{l+1}\right)\right)+{\hat{g}}_{l+1}$$

In the above equation, ${\hat{g}}_{l}$ and $g_{l}$, respectively, denote the outputs of the (S)W-MSA [21] module and the MLP module for the $l^{th}$ block. The computation of self-attention is as follows:(18)$$Attention\left(Q,K,V\right)=Softmax\left(\frac{QK^{T}}{\sqrt{d}}+B\right)V$$
where in Equation (Equation 18), $Q,K,V\in R^{M^{2}\times d}$ represent the query, key, and value matrices, respectively. Here, $M^{2}$ and *d* refer to the number of patches in the window and the dimension of the query or key. The elements of matrix *B* are sourced from the bias matrix $\hat{B}\in R^{\left(2M-1)\times (2M+1\right)}$.

Following the feature extraction by Swin Transformer blocks, the feature maps are divided into multiple small patches. As shown in Figure 3b, the SwinTFAM processes these patches, implementing a downsampling operation by selecting elements at intervals of two in both row and column directions, consequently reducing the height and width of the image by half. Subsequently, the selected elements are reorganized into new patches. All these newly formed patches are concatenated to form a unified tensor. In this process, due to the reduction in the size of the image, the number of channels is increased to four times the original amount. A fully connected layer then adjusts the dimensions of this tensor, reducing it to twice the original size. This sequence of operations not only reduces the resolution of the image but also expands the number of channels, facilitating the model in capturing more abstract levels of features.

The processing flow of the SwinTFRM is shown in Figure 3c. After the feature map $X\in R^{H\times W\times C}$ is extracted by the Swin Transformer, it first passes through a convolutional layer with a PReLU activation function, keeping the dimension as H×W×C. Then, it is upsampled using bilinear interpolation to 2H×2W×C, followed by a convolutional layer that reduces the number of channels to C/2. Simultaneously, *X* is fed into a convolutional layer with a kernel size of 1×1, doubling the number of channels to 2C. Afterward, a pixel shuffle operation is applied, reshaping the dimensions to 2H×2W×C/2. The two upsampled results are then fused along the channel dimension, forming the final feature map with dimensions 2H×2W×C. Lastly, a convolutional layer reduces the number of channels by half, producing the final upsampled feature map with dimensions 2H×2W×C/2.

The entire operation of the FR module can be expressed by the following equation:(19)$$X_{out}=\mathrm{Conv}\left(\mathrm{Concat}\left(\mathrm{Bilinear}(\mathrm{Conv}(X)),\mathrm{PixelShuffle}(\mathrm{Conv}(X))\right)\right)$$
where Conv(X) represents the combination of convolution and PReLU activation function; Bilinear(·) denotes bilinear interpolation for upsampling; PixelShuffle(·) represents the pixel shuffle operation; and Concat(·) is the channel fusion operation. The final Conv(·) is the convolutional operation that reduces the number of channels by half. This process effectively enhances the resolution while retaining rich spectral information.

## 4. Experiments and Analyses

### 4.1. Datasets

In this study, we utilized three widely recognized hyperspectral image benchmark datasets: CAVE [46], Washington DC Mall (WDCM) [47], and Pavia University (PU) [48], along with a real remote sensing dataset, ZY-1E.

The CAVE dataset consists of 32 indoor hyperspectral images, each with a resolution of 512 × 512 pixels and spanning 31 spectral bands in the 400–700 nm wavelength range. Following the Wald protocol [49], HR-MSI was generated using the Nikon D700 camera’s spectral response function. The original hyperspectral images served as the reference HR-HSI. To create LR-HSI, the HR-HSI images were first blurred using an 8 × 8 Gaussian kernel with a standard deviation of 2, followed by downsampling by a factor of eight in both spatial dimensions. For training, 20 image pairs were randomly selected, leaving the remaining 12 pairs for testing. During training, 64 × 64 patches were randomly cropped from the 512 × 512 images, resulting in patch sizes of 64 × 64 × 31 for HR-HSI, 64 × 64 × 3 for HR-MSI, and 8 × 8 × 31 for LR-HSI. In the testing phase, the full-sized images were used, with dimensions of 512 × 512 × 31 for HR-HSI, 512 × 512 × 3 for HR-MSI, and 64 × 64 × 31 for LR-HSI.

The WDCM dataset, captured by the HYDICE sensor in 1995, comprises 191 spectral bands spanning the 400–2400 nm wavelength range, with each band having a spatial resolution of 1280 × 307 pixels at 2.5 m per pixel. Two sub-images, each 128 × 128 pixels, were extracted for testing, while the remaining data were used for training. Following the same setup as MSST-Net [40], HR-MSI was generated using the spectral response matrix of the Sentinel-2A sensor, and LR-HSI was created in the same manner as in the CAVE dataset. During training, the dimensions of HR-HSI, HR-MSI, and LR-HSI were 64 × 64 × 191, 64 × 64 × 10, and 8 × 8 × 191, respectively, while for testing, they were 128 × 128 × 191, 128 × 128 × 10, and 16 × 16 × 191.

The PU dataset, collected by the ROSIS sensor in 2003, originally measured 610 × 340 pixels and covered a wavelength range of 430–860 nm. After removing 22 bands affected by water vapor absorption, 93 spectral bands remained. Similar to the WDCM dataset, LR-HSI was generated by applying Gaussian filtering followed by downsampling by a factor of eight. Two sub-images of 128 × 128 pixels were reserved for testing, while the rest were used for training. HR-MSI was produced using a spectral response function comparable to the IKONOS sensor. The dimensions of HR-HSI, HR-MSI, and LR-HSI during training were 64 × 64 × 93, 64 × 64 × 4, and 8 × 8 × 93, respectively, while for testing, they were 128 × 128 × 93, 128 × 128 × 4, and 16 × 16 × 93.

The ZY-1E dataset used in this study was acquired from the ZY-1E satellite. The satellite carries two payloads, namely a visible near-infrared camera and a hyperspectral camera. We obtained a hyperspectral and multispectral image captured on 19 April 2023, in the Pinggu district of Beijing, China. For the hyperspectral image and multispectral image, we performed preprocessing steps such as radiometric calibration, atmospheric correction, and orthorectification in ENVI software (version 5.3) [50]. We then registered the hyperspectral image with the multispectral image [51] using the ’Georeferencing’ tool in ArcGIS, where 20 control points were manually added and the first-order polynomial transformation method was selected to ensure residuals were less than 1, resulting in a final average residual error of 0.6. The original dataset consists of an LR-HSI with 166 bands and an HR-MSI with 8 bands. The spatial resolutions of LR-HSI and HR-MSI are 30 m and 10 m, respectively, with corresponding spatial sizes of 4986 × 4581 and 1662 × 1527. It is worth noting that due to the lack of a reference image for HR-HSI, we performed spatial downsampling on the original LR-HSI and HR-MSI using a Gaussian kernel of size 9 × 9, following the Wald protocol, to generate the training set. We then treated the original LR-HSI as HR-HSI. Due to the relatively low signal-to-noise ratio in some bands of the ZY-1E data, we selected the first 76 bands of LR-HSI for fusion experiments. During the training phase, we randomly cropped image pairs (HR-HSI: 60 × 60 × 76, HR-MSI: 60 × 60 × 8, LR-HSI: 20 × 20 × 76) from the training set for training. In the testing phase, we cropped image pairs (HR-MSI: 540 × 540 × 8, LR-HSI: 180 × 180 × 76) for testing purposes.

### 4.2. Implementation and Indices

#### 4.2.1. Implementation Details

In our experiments, for the Swin Transformer Block, we configured the parameters as follows: window size of 8, number of channels set to 96, attention heads numbered at 8, and a secondary window size of 4. The Adam optimizer was utilized for a total of 625,000 iterations, divided into 100 epochs with 625 iterations each. The initial learning rate was set to 0.0001, with the rate decaying by a factor of 2 at the 50th and 80th epochs. The batch size was established at 32. Our network model was implemented and trained on a Linux operating system (Ubuntu 20.04) using PyTorch 1.11.0, Python 3.8, and CUDA 11.3.

#### 4.2.2. Evaluation Indices

To comprehensively evaluate the performance of image fusion algorithms, a suite of quantitative indices is commonly employed. These include the peak signal-to-noise ratio (PSNR), spectral angle mapper (SAM), Erreur Relative Globale Adimensionnelle de Synthèse (ERGAS), structural similarity index measure (SSIM), root mean square error (RMSE), and the quality with no reference (QNR) index. PSNR assesses spatial effectiveness, with higher values indicating lesser spatial information loss. SAM evaluates spectral quality, with lower values denoting reduced spectral information loss. ERGAS, an overall assessment metric, indicates better fusion quality with lower values. SSIM gauges structural correlation, where higher values reflect better fusion effectiveness. RMSE measures similarity between images, with lower values suggesting superior algorithm performance. QNR is particularly well-suited for assessing the fusion quality of no-reference imagery, such as the ZY-1E real remote sensing dataset, as it captures both spectral and spatial distortions. Higher QNR values indicate better fusion quality, reflecting more effective preservation of information. Together, these metrics evaluate the ability of fusion algorithms to retain both spatial and spectral details.

### 4.3. Ablation Experiments

To further investigate the roles of the main modules in our proposed network, we conducted a series of ablation experiments on the CAVE dataset using the FCSwinU model as a baseline. Specifically, to verify the superiority of the FFT module in the SFFC for shallow feature extraction compared to other modules, we replaced the FFT module with a commonly used convolutional residual connection structure for extracting shallow LR-HSI features. The experimental results are shown in the second column of Table 1. Compared to the baseline model, the absence of the FFT structure resulted in a decrease of 0.75 dB in PSNR, as well as a decrease in all other performance metrics. This result indicates that pre-extracting low-resolution hyperspectral features using the FFT structure in the SFFC module is highly beneficial for improving image fusion performance.

The encoder–decoder (ED) structure has been widely employed in the process of extracting spectral and spatial information due to its strong feature propagation capability and advantages in recovering details. In our model, skip connections are designed to directly transfer low-level features to the decoder part, enabling the decoder to effectively utilize rich low-level feature information. To validate the effectiveness of the ED structure, we conducted an ablation experiment where the ED structure was removed, and only the SwinT was used to extract deep features. The experimental results are presented in the third column of Table 1. It is evident from the results that the performance without the ED structure is significantly lower than the baseline model across all evaluation metrics. This finding suggests that the ED structure plays a crucial role in extracting spectral features by facilitating multi-scale feature extraction.

The self-attention mechanism within the Swin Transformer enables the extraction of global features, which, compared to traditional CNN that extracts local features, more efficiently achieves the fusion of hyperspectral and multispectral images. In our FCSwinU model, we fully leverage the windowed self-attention mechanism of the Swin Transformer to handle HR-MSI and LR-HSI, in order to explore their spatial and spectral features more comprehensively. To validate the effectiveness of the Swin Transformer, we conducted an ablation experiment where only the encoder–decoder architecture was used, and a CNN was employed as a replacement for the SwinT. It is evident from the results in the fourth column of Table 1 that all performance metrics are significantly lower than the baseline. This finding indicates that without the SwinT, the ability of our proposed network to extract deep features would be greatly diminished.

In addition to this, we further replaced the FFT module with a Temporal Convolutional Network (TCN), as shown in the last column of Table 1. This experiment was conducted to compare the performance of FFT-based frequency domain convolution and TCN-based real threshold convolution for shallow feature extraction. The results show that replacing FFT with TCN leads to a further decline in performance across all metrics, especially in PSNR and SSIM. This confirms that FFT-based shallow feature extraction in the SFFC module provides a more effective way to capture detailed spectral information, which is crucial for improving the accuracy and quality of hyperspectral image reconstruction.

In summary, this study presents an efficient network for the fusion of hyperspectral and multispectral images. The network synergistically integrates the windowed self-attention mechanism of the Swin Transformer with the potent feature extraction capabilities of the SFFC module, maximizing the utilization of the interrelated spatial and spectral features. In the encoder, the windowed self-attention mechanism precisely extracts feature representations of HR-MSI and LR-HSI, effectively capturing long-range dependencies and global information within the images. The decoder employs multi-scale transformations and feature fusion techniques to reconstruct and merge these features, resulting in high-quality fused images. The network’s exceptional fusion performance in hyperspectral and multispectral image fusion has been thoroughly validated and exhibited through a series of ablation experiments.

### 4.4. Results and Analysis

#### 4.4.1. Experiments with Benchmark Data

In this section, our proposed fusion approach is compared with eight other state-of-the-art methods on the CAVE, WDCM, and PU datasets. These comparative methods include three traditional techniques, such as the unmixing-based HySure [52] and CNMF [53] algorithms, and the multi-resolution analysis-based GLP-HS [47] algorithm, along with five recent deep learning-based techniques, namely SSFCNN [28], MSDCNN [54], MHF-Net [14], Fusformer [19], and MSST-Net [40]. Among these, Fusformer and MSST-Net are Transformer-based fusion methods, while the other three employ CNN-based fusion techniques.

Table 2, Table 3 and Table 4, respectively, present the average quantitative results on the CAVE, WDCM, and PU test sets, highlighting the best results in bold. It is observed that CNN-based methods performed well, outperforming traditional methods overall, attributed to the excellent non-linear feature extraction capability of CNNs. MSST-Net achieved results close to ours due to its multi-scale training in both spectral and spatial dimensions, followed by the Fusformer method. All Transformer-based methods demonstrated considerable fusion performance across these three datasets, indicating significant advantages in extracting global features for Transformer-based methods. Our approach, FCSwinU, exhibited the best performance across PSNR, SAM, ERGAS, SSIM, and RMSE metrics on these three datasets. Particularly notable is our method’s achievement in global spatial fusion PSNR metrics, reaching 47.58, 50.73, and 42.91, respectively. Leveraging the SFFC module for extracting features from LR-HSI, our method, unlike Fusformer and MSST-Net, utilizes the SwinTFAM and SwinTFRM to enhance feature fusion and extraction from different modalities.

It is worth noting that our method performs comparably to the recently proposed MSST-Net in terms of performance, mainly because both approaches employ attention mechanisms and multi-scale feature extraction. However, our method differs from MSST-Net in that we utilize a window-based attention mechanism, whereas MSST-Net relies solely on pure attention mechanisms. As mentioned in our introduction section, research suggests that window-based attention mechanisms have advantages over pure multi-head attention mechanisms. Additionally, we introduce the use of the fast Fourier transform to extract shallow features from hyperspectral images, whereas MSST-Net only employs CNN residual connections for shallow feature extraction.

To evaluate the computational efficiency, the last two columns of Table 2 detail the floating-point operations (FLOPs) and parameter counts for various fusion methods on the CAVE dataset. As indicated, Fusformer and MSST-Net exhibit notably higher computational demands, with FLOPs of 456.327 G and 188.720 G, respectively. Fusformer’s use of standard Transformer layers likely contributes to this heavy load, while MSST-Net’s high parameter count of 34.400 M may stem from the use of large-scale transposed convolutions. In contrast, methods such as SSCNN and MSDCNN maintain more balanced computational profiles, featuring moderate FLOPs and parameter counts. For example, SSCNN operates with 1.727 G FLOPs and 0.423 M parameters. Our proposed method, FCSwinU, achieves an optimal trade-off, balancing performance with computational efficiency, registering 1.125 G FLOPs and a parameter count of 29.176 M, making it efficient for practical applications without sacrificing fusion quality.

In order to visually assess the quality of the fused images, we present SAM maps and error maps for some bands and their corresponding fusion results (shown in Figure 4, Figure 5, Figure 6, Figure 7, Figure 8 and Figure 9). Figure 4 and Figure 5 exhibit false-color images synthesized from three estimated bands randomly selected from the CAVE test set, along with their corresponding error maps. For comparative clarity, the second row in all result images shows SAM images of spectral effects produced by different fusion methods, displayed using the ’jet’ color map. Colors closer to deep blue indicate smaller SAM values, corresponding to lower spectral information loss, while colors closer to red signify larger SAM values, indicating greater spectral information loss. The third row in the images illustrates error images between the fusion results of various spectral fusion methods and the ground truth reference images. By subtracting the fusion results from the reference images and averaging over bands, displayed using the ’summer’ color map, we map the images to a deep green color. Consequently, as the error images approach deep green, the errors diminish, indicating better fusion performance.

On the WDCM dataset, we present pseudo-color images synthesized from the 67th, 27th, and 17th estimated bands, along with their corresponding SAM maps and error maps (Figure 6 and Figure 7). It can be observed that all deep learning methods produced competitive results. However, our method demonstrates superior quality results, with fewer residuals evident in the error maps, particularly in the region highlighted by the red box. In Figure 8 and Figure 9, we display pseudo-color images synthesized from the 29th, 19th, and 9th estimated bands of the PU dataset, accompanied by their respective SAM maps and error maps. Apart from our method, none of the other approaches managed to achieve satisfactory results. A challenge in image reconstruction lies in restoring texture features. Our method excels in recovering spatial textures, prominently observable in regions rich in texture, such as the area highlighted by the red box.

#### 4.4.2. PSNR Comparison across Spectral Bands

We compared and analyzed the average PSNR of various methods across different spectral bands in multiple datasets. For each dataset, a fusion image was randomly selected from the test set, and its PSNR across various bands was calculated and averaged to obtain the corresponding PSNR curve for each band. Figure 10a,c, respectively, illustrate the average PSNR curves of different methods across various bands in the CAVE and PU datasets. These figures reveal that, although the overall performance trends of the methods are similar across different bands, indicating effective capture of fundamental band information, our FCSwinU method typically ranks at the top of these curves, demonstrating superior detail control capability. Figure 10b presents the experimental results on the WDCM dataset. Unlike the CAVE and PU datasets, the spectral bands of WDCM are more complex, with richer geomorphological information reflected in each band. In analyzing the PSNR curves of the WDCM dataset, the FCSwinU algorithm exhibits its powerful capacity to handle complex terrain information. Notably, in parts of the PSNR curve where sharp declines and rises occur, FCSwinU usually recovers more rapidly, indicating its swift adaptability to sudden changes in image quality. This finding suggests that FCSwinU not only excels in simple scenarios but also effectively captures and reproduces the details and features of terrain, even in the more informationally rich and complex WDCM dataset. This proficiency is attributed to its advantages in multi-scale feature extraction, deep feature learning, and efficient information integration strategies.

Through a comprehensive comparison of fusion results obtained from various methods, it is evident that images derived from traditional methods (HySure, CNMF, and GLP-HS) exhibit noticeable color differences and spatial distortions, particularly in enlarged regions. In contrast, our proposed deep learning-based method, FCSwinU, demonstrates clearer local structural information and reduced spectral distortion. Our approach exhibits minimal errors at image edges and corner positions, showcasing superior spectral quality with minimal spectral information loss, thereby validating the outstanding performance of FCSwinU in reconstructing fine details of hyperspectral images. Despite the ability of deep learning-based methods to generate estimated bands with similar visual effects, our method presents fewer residues in error maps, especially in the region marked by the red box, indicating our enhanced preservation of spatial details in the estimated bands.

#### 4.4.3. Experiments with Real Data

To validate the robustness of our method on real data, we conducted tests using the real LR-HSI and HR-MSI dataset, ZY-1E. During the testing process, due to limitations in GPU memory, we cropped the LR-HSI of the test images to a size of 180 × 180, corresponding to a cropped size of 540 × 540 for the HR-MSI. In order to better observe the fusion results, Figure 11 displays the reconstructed images containing various scenes such as water bodies and buildings. A comparison with the LR-HSI reveals varying degrees of color distortion in the images produced by MHF-Net, and Fusformer, while HySure and CNMF exhibit severe distortion and spatial aliasing. The images reconstructed by SSFCNN and MSDCNN perform worse than other deep learning-based methods in terms of spatial details. The image produced by FCSwinU closely resembles the color of the LR-HSI and enhances spatial details. Compared to other methods, FCSwinU demonstrates the best performance.

Figure 12 presents the spectral curves of various objects from the ZY-1E dataset after fusion using different methods, compared to the original LR-HSI. The examples include (a) playgrounds and (b) cultivated land. In these plots, the spectral curves of our proposed FCSwinU are shown in red, while the reference LR-HSI curves are in black. Notably, the spectral curves of the FCSwinU, with slight variations, closely align with those of the LR-HSI, reflecting minimal error across all objects. In contrast, the traditional HySure method performs the worst, with significant deviations, particularly in the playground case. MSST-Net performs better, with smaller spectral errors in the same scenario. Furthermore, we utilized the QNR metric to assess the quality of the fused images. As indicated in Table 5, FCSwinU achieved the highest QNR score, highlighting its superior performance in image fusion quality compared to other methods.

## 5. Conclusions

In this study, we introduce FCSwinU, an innovative network for hyperspectral and multispectral image fusion. By combining the Swin Transformer’s windowed self-attention mechanism with a UNet-like encoder–decoder structure, FCSwinU effectively processes HR-MSI and LR-HSI data. The network features the SFFC module for early LR-HSI feature extraction, alongside the SwinTFAM and SwinTFRM modules for feature fusion and reconstruction. Experimental results show that FCSwinU delivers superior performance in image fusion tasks.

However, we recognize that upsampling LR-HSI for feature extraction and fusion may result in some loss of spatial information. As a limitation of the current approach, this could impact the overall fusion quality. Future work will focus on exploring alternative upsampling methods, such as transposed convolution, to better preserve spatial details and improve alignment between HR-MSI and LR-HSI. This enhancement is critical for addressing real-world challenges, such as variations in pose and lighting conditions, thereby increasing the robustness and generalization of FCSwinU across diverse applications.

Moreover, while the proposed network demonstrates excellent fusion performance, its generalization across different datasets remains limited due to variations in spectral bands and image characteristics. Addressing this issue is another key future direction, where we aim to explore solutions that improve the model’s adaptability and enable its broader applicability across datasets.

## Figures and Tables

**Figure 1 sensors-24-07023-f001:**
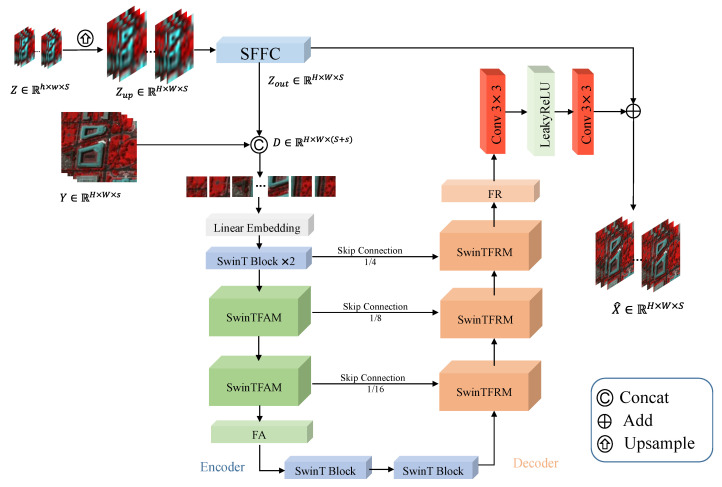
The overall architecture diagram of the proposed FCSwinU network.

**Figure 2 sensors-24-07023-f002:**
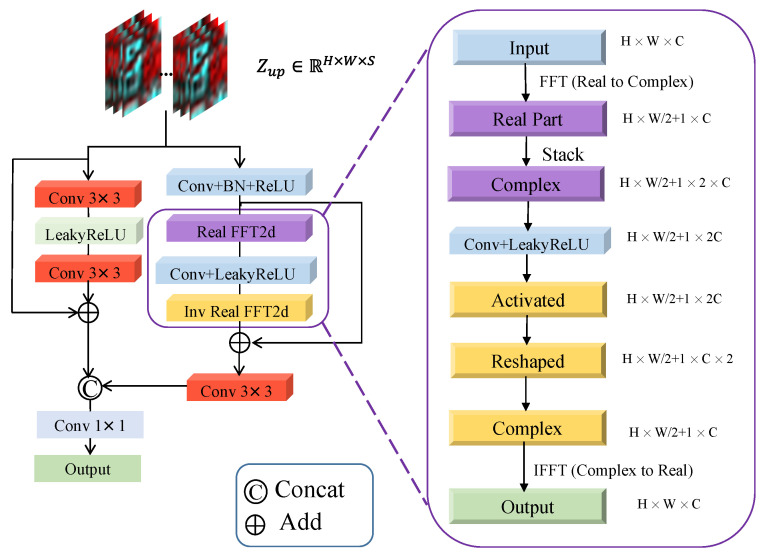
The architecture of the SFFC module.

**Figure 3 sensors-24-07023-f003:**
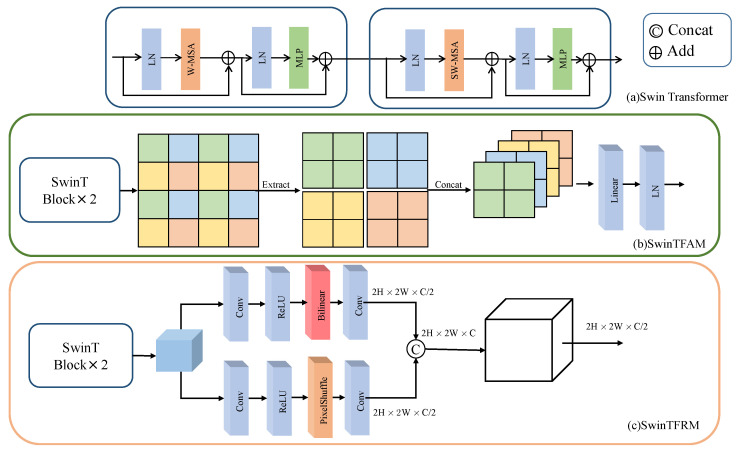
Schematic diagram of the Swin Transformer feature abstraction and reconstruction module: (**a**) represents the structure of the Swin Transformer; (**b**) represents the Swin Transformer feature abstraction module; (**c**) represents the Swin Transformer feature reconstruction module.

**Figure 4 sensors-24-07023-f004:**
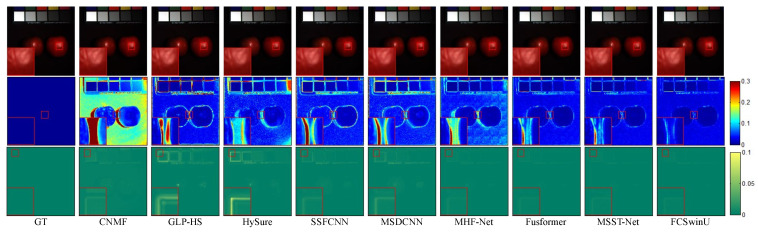
Illustrates the fusion results of the “fake and real apple” image from the CAVE dataset. The first row presents a false-color image synthesized from the 29th, 19th, and 9th spectral bands. The second row visualizes the SAM map. The third row depicts the error images between the fused and the actual images.

**Figure 5 sensors-24-07023-f005:**
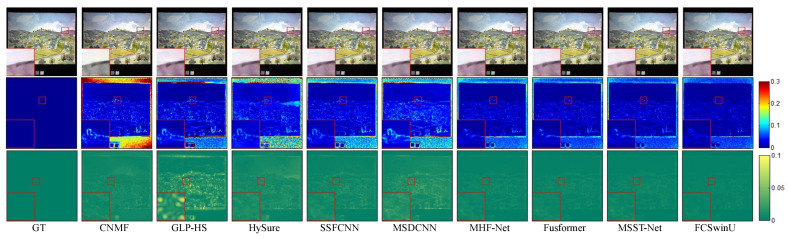
Fusion results of the “watercolors” Image from the CAVE Dataset. The first row presents a false-color image synthesized from the 67th, 27th, and 17th spectral bands. The second row visualizes the SAM map. The third row depicts the error images between the fused and the original images.

**Figure 6 sensors-24-07023-f006:**
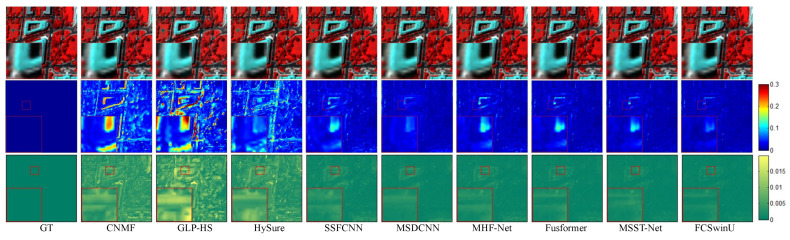
Fusion results of Image 1 from the WDCM test set. The first row presents a false-color image synthesized from the 67th, 27th, and 17th spectral bands. The second row visualizes the SAM map. The third row depicts the error images between the fused and the original images.

**Figure 7 sensors-24-07023-f007:**
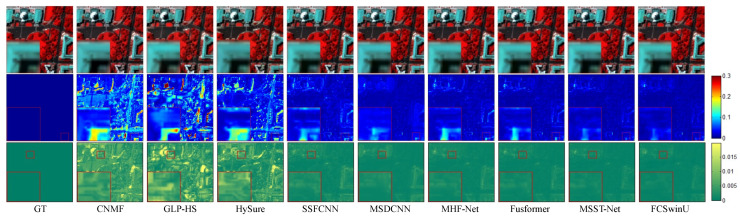
Fusion results of Image 2 from the WDCM test set. The first row presents a false-color image synthesized from the 67th, 27th, and 17th spectral bands. The second row visualizes the SAM map. The third row depicts the error comparison between the fused and the original images.

**Figure 8 sensors-24-07023-f008:**
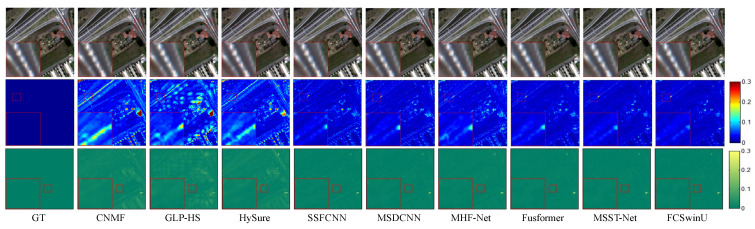
Fusion results of Image 1 from the PU test set. The first row presents a false-color image synthesized from the 29th, 19th, and 9th spectral bands. The second row visualizes the SAM map. The third row depicts the error comparison between the fused and the original images.

**Figure 9 sensors-24-07023-f009:**
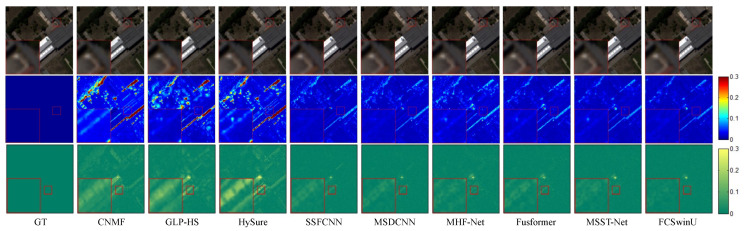
Fusion results of Image 2 from the PU test set. The first row presents a false-color image synthesized from the 29th, 19th, and 9th spectral bands. The second row visualizes the SAM map. The third row depicts the error comparison between the fused and the original images.

**Figure 10 sensors-24-07023-f010:**
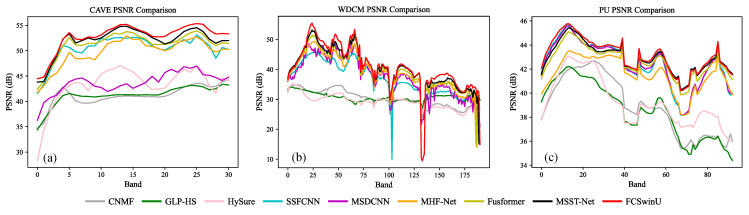
Average PSNR comparison of various fusion algorithms across different spectral bands in (**a**) CAVE, (**b**) WDCM, and (**c**) PU datasets.

**Figure 11 sensors-24-07023-f011:**
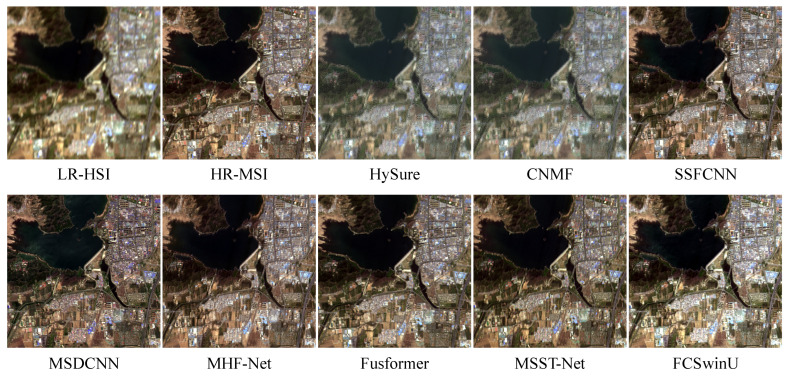
Qualitative result 1 of the ZY-1E dataset. We show the composite image of the MSI with bands 3-2-1 as R-G-B and HSI with bands 29-19-10 as R-G-B. reconstructed images by 7 comparison methods.

**Figure 12 sensors-24-07023-f012:**
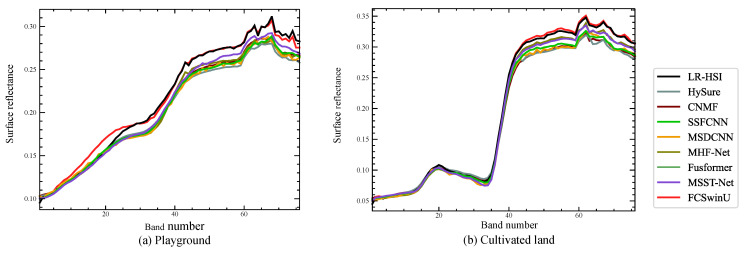
Spectral contrast of different objects in the ZY-1E dataset.

**Table 1 sensors-24-07023-t001:** Ablation experimental results on the CAVE dataset.

	FFT	SwinT	ED	Base	TCN
FFT	✗	✓	✓	✓	✗
SwinT	✓	✗	✓	✓	✓
ED	✓	✓	✗	✓	✓
base	✓	✓	✓	✓	✓
PSNR ↑	46.83	45.42	46.41	**47.58**	47.56
SAM ↓	1.45	1.67	1.49	**1.34**	1.24
SSIM ↑	0.9947	0.9935	0.9945	**0.9955**	0.9947
RMSE ↓	0.0046	0.0054	0.0047	**0.0042**	0.0046
ERGAS ↓	1.10	1.28	1.14	**1.02**	1.07

Bold values indicate the best performance for each metric. ↑ indicates that a higher value is better, while ↓ indicates that a lower value is better.

**Table 2 sensors-24-07023-t002:** Quantitative evaluation of new and existing methods across seven metrics in the CAVE Dataset.

Method	PSNR ↑	SAM ↓	ERGAS ↓	SSIM ↑	RMSE ↓	FLOPs (G)	Para (M)
CNMF [53]	37.95	3.35	3.16	0.9762	0.0157	/	/
GLP-HS [47]	36.50	3.68	3.10	0.9729	0.0196	/	/
HySure [52]	40.39	2.61	2.62	0.9832	0.0096	/	/
SSFCNN [28]	41.87	2.43	1.84	0.9857	0.0078	1.727	0.423
MSDCNN [54]	39.84	3.04	2.90	0.9775	0.0100	2.209	0.527
MHF-Net [14]	45.93	1.58	1.21	0.9934	0.0050	22.460	3.630
Fusformer [19]	46.74	1.45	1.11	0.9945	0.0046	456.327	0.109
MSST-Net [40]	47.16	1.39	1.06	0.9950	0.0044	188.720	34.400
**FCSwinU**	**47.58**	**1.34**	**1.02**	**0.9955**	**0.0042**	1.125	29.176

Bold values indicate the best performance for each metric. ↑ indicates that a higher value is better, while ↓ indicates that a lower value is better.

**Table 3 sensors-24-07023-t003:** Quantitative evaluation of new and existing methods across five metrics in the WDCM Dataset.

Method	PSNR ↑	SAM ↓	ERGAS ↓	SSIM ↑	RMSE ↓
CNMF [53]	37.28	4.10	2.93	0.9938	0.0077
GLP-HS [47]	34.31	5.37	3.81	0.9897	0.0095
HySure [52]	35.65	4.74	3.52	0.9915	0.0094
SSFCNN [28]	45.50	1.32	0.90	0.9896	0.0023
MSDCNN [54]	45.90	1.28	0.86	0.9986	0.0022
MHF-Net [14]	48.22	1.24	0.83	0.9985	0.0021
Fusformer [19]	49.24	1.10	0.74	0.9989	0.0019
MSST-Net [40]	49.98	1.01	0.67	0.9990	0.0018
**FCSwinU**	**50.73**	**0.93**	**0.63**	**0.9991**	**0.0016**

Bold values indicate the best performance for each metric. ↑ indicates that a higher value is better, while ↓ indicates that a lower value is better.

**Table 4 sensors-24-07023-t004:** Quantitative evaluation of new and existing methods across five metrics in the PU Dataset.

Method	PSNR ↑	SAM ↓	ERGAS ↓	SSIM ↑	RMSE ↓
CNMF [53]	34.73	5.07	2.91	0.9716	0.0309
GLP-HS [47]	35.95	4.70	2.48	0.9753	0.0276
HySure [52]	36.05	4.16	2.30	0.9747	0.0256
SSFCNN [28]	41.08	2.43	1.34	0.8933	0.0223
MSDCNN [54]	42.07	2.24	1.24	0.9885	0.0125
MHF-Net [14]	41.19	2.48	1.23	0.9841	0.0139
Fusformer [19]	42.22	2.20	1.12	0.9897	0.0124
MSST-Net [40]	42.61	2.11	1.08	0.9910	0.0118
**FCSwinU**	**42.91**	**2.04**	**1.04**	**0.9916**	**0.0114**

Bold values indicate the best performance for each metric. ↑ indicates that a higher value is better, while ↓ indicates that a lower value is better.

**Table 5 sensors-24-07023-t005:** No-reference indexes for the fusion results of each method on the ZY-1E dataset. (The higher, the better).

Method	HySure	CNMF	SSFCNN	MSDCNN	MHF-Net	Fusformer	MSST-Net	FCSwinU
QNR	0.932	0.934	0.958	0.957	0.961	0.965	0.966	**0.968**

## Data Availability

The publicly available dataset used in the experiments mentioned in the article can be accessed via the following link: https://github.com/meiruni/MIMFormer. Additionally, the source code is available at: https://github.com/meiruni/FCSwinU.

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
