# Peer review of "FCSwinU: Fourier Convolutions and Swin Transformer UNet for Hyperspectral and Multispectral Image Fusion"

_sensors, 2024, doi:10.3390/s24217023_

Round 1
Reviewer 1 Report
Comments and Suggestions for Authors
This paper proposed a new deep learning network architecture called FCSwinU, which is used to fuse low spatial resolution hyperspectral images (LR-HSI) and high spatial resolution multispectral images (HR-MSI) to obtain high spatial resolution hyperspectral images (HR-HSI). The proposed method performs well in considering multi-scale and spectral resolution differences. It not only proposes a novel theoretical method, but also verifies the effectiveness of the method on a real dataset through experiments. However, the image registration problem has not been explored in depth, and the generalization ability of the model needs to be improved:
Just some of the things that need improvement:
(1) The introduction is too long. It is recommended to reduce the content to improve readability.
(2) The logic of the first three paragraphs of the introduction is not clear enough. What is the focus of the article? It is recommended to revise it carefully.
(3) In Figure 1, the input should be clear low-spatial-resolution hyperspectral image and high-spatial-resolution multispectral image (HR-MSI). Has the low-resolution image been sampled? Since it is multi-source data fusion, why is sampling required?
(4) Line 400, list references.
(5) What is the registration error between hyperspectral images and multispectral images? How are they registered? This needs to be explained in the text.
(6) When applied to real ZY data, the difference is not big as can be seen from the figure. This part seems to lack the fusion of different models on ZY data, and also lacks the error comparison between the fused image and the original image.
Comments on the Quality of English Language
English language is good
Author Response
Dear reviewer,
Thank you very much for the opportunity to improve the quality of our manuscript, "FCSwinU:Fourier Convolutions and Swin Transformer Unet for Hyperspectral and Multispectral Image Fusion" (sensors-3145225). We greatly appreciate the constructive and insightful comments from you.
We have carefully revised the manuscript based on your comments. We hope that the revised manuscript now meets the publication standards of the Sensors.
To facilitate the review process, revisions in the manuscript are highlighted using yellow, green, and red. Yellow indicates minor changes, green denotes significant modifications, and red marks major revisions. Additionally, our point-by-point responses to your comments are provided below.
Comment 1:The introduction is too long. It is recommended to reduce the content to improve readability.
Response: Thank you very much for your valuable feedback. We sincerely appreciate your suggestion and have now shortened the introduction to improve its readability.
Comment 2:The logic of the first three paragraphs of the introduction is not clear enough. What is the focus of the article? It is recommended to revise it carefully.
Response: We are grateful for your insightful comment regarding the logical flow of the first three paragraphs. To address this, we have carefully revised these sections to better highlight the article's focus and provide a clearer narrative.
Comment 3:In Figure 1, the input should be clear low-spatial-resolution hyperspectral image and high-spatial-resolution multispectral image (HR-MSI). Has the low-resolution image been sampled? Since it is multi-source data fusion, why is sampling required?
Response: Thank you for your insightful comment. We would like to clarify the reasoning behind the sampling operation in our architecture. Although this is a multi-source data fusion task, upsampling is necessary for the following reasons:
- Spatial Resolution Alignment: The LR-HSI needs to be upsampled to match the HR-MSI's spatial resolution for effective feature matching and fusion.
- Effective Feature Matching: Upsampling ensures that the spatial scales of both data types are aligned, enabling better fusion and extraction of complementary information.
- Final Output Resolution: Upsampling allows the network to recover high-resolution spatial information in the final HR-HSI output while preserving the spectral richness of LR-HSI.
In summary, upsampling is essential for aligning spatial resolutions, facilitating effective fusion, and achieving the desired high-resolution output.
Comment 4:Line 400, list references.
Response: Thank you for your valuable feedback. We have listed the references in line 400 of the text based on your suggestions. We appreciate your attention to detail and thank you for your constructive feedback.
Comment 5:What is the registration error between hyperspectral images and multispectral images? How are they registered? This needs to be explained in the text.
Response: Thank you for your valuable comments. The registration process was conducted using the "Georeferencing" tool in ArcGIS. We manually added 20 control points and selected the first-order polynomial transformation method to ensure that the residuals were less than 1. The final average residual error was 0.6. Additionally, we visually inspected the alignment by zooming in on the boundaries to confirm proper registration. We have now added this explanation to the data introduction section, as per your suggestion. Thank you once again for highlighting this aspect of the image alignment process!
Comment 6:When applied to real ZY data, the difference is not big as can be seen from the figure. This part seems to lack the fusion of different models on ZY data, and also lacks the error comparison between the fused image and the original image.
Response: Thank you for your valuable comment. Considering that the differences are relatively minor, we have added spectral curve comparison graphs between the fused results of various methods and the original LR-HSI data in the Playground and Cultivated land regions. Additionally, we acknowledge that the omission of error comparisons between the fused images and the original data in the initial manuscript was an oversight. In the revised version, we have included detailed results, including the QNR quantitative evaluation metrics, to thoroughly demonstrate the performance of our method in real-world applications. We appreciate your insightful feedback, which has significantly enhanced the practical value and academic contribution of our work.

Reviewer 2 Report
Comments and Suggestions for Authors
This article uses Fourier Convolution and Swin Transformer techniques to design an HSI and MSI fusion method, in general, I have the following concerns about this paper:
1. The INTRODUCTION section should contain knowledge of hyperspectral imaging, not just its applications. For example, doi:10.1109/TGRS.2024.3361929.
2. the authors use Fourier frequency domain convolution instead of the common time domain convolution, then the authors should provide objective experimental evidence that this is effective.
3. The HR-MSI and LR-HSI data are unclear about the construction process and it is recommended that this part of the description be scrutinized.
4. In the section on ablation experiments, it is recommended that the authors give performance indicators for different scales.
5. In addition to the spatial error assessment, it is recommended that the authors give the spectral reconstruction errors.
6. It is recommended that parametric quantities and FLOPs for all methods be presented.
7. The concluding section should contain the limitations of the proposed method.
Comments on the Quality of English Language
Moderate editing of English language required.
Author Response
Dear reviewer,
Thank you very much for the opportunity to improve the quality of our manuscript, "FCSwinU:Fourier Convolutions and Swin Transformer Unet for Hyperspectral and Multispectral Image Fusion" (sensors-3145225). We greatly appreciate the constructive and insightful comments from you.
We have carefully revised the manuscript based on your comments. We hope that the revised manuscript now meets the publication standards of the Sensors.
To facilitate the review process, revisions in the manuscript are highlighted using yellow, green, and red. Yellow indicates minor changes, green denotes significant modifications, and red marks major revisions. Additionally, our point-by-point responses to your comments are provided below.
Comment 1:The INTRODUCTION section should contain knowledge of hyperspectral imaging, not just its applications. For example, doi:10.1109/TGRS.2024.3361929.
Response: Thank you for your valuable feedback. We appreciate your suggestion to include more foundational knowledge of hyperspectral imaging (HSI) in the Introduction section. In response, we have revised the first paragraph to provide a more comprehensive overview of HSI, not only emphasizing its applications but also explaining its underlying principles and importance in material identification across various fields. This addition aims to enhance the reader's understanding of the significance of HSI before discussing its challenges and potential solutions.
Comment 2:the authors use Fourier frequency domain convolution instead of the common time domain convolution, then the authors should provide objective experimental evidence that this is effective.
Response: Thank you for your valuable comment. We have added detailed evidence in the ablation study to demonstrate the effectiveness of using Fourier frequency domain convolution. Specifically, we replaced it with standard time-domain convolution for comparison, and the results confirm the advantages of our approach.
Comment 3:The HR-MSI and LR-HSI data are unclear about the construction process and it is recommended that this part of the description be scrutinized.
Response: Thank you for your valuable feedback. We have carefully reviewed and revised the section describing the construction process of the high-resolution multispectral imaging (HR-MSI) and low-resolution hyperspectral imaging (LR-HSI) data. In the updated manuscript, we have provided a more detailed explanation of how these datasets were constructed and utilized in our study. We appreciate your attention to detail, which has greatly helped us improve the clarity and overall quality of the manuscript.
Comment 4:In the section on ablation experiments, it is recommended that the authors give performance indicators for different scales.
Response: Thank you very much for your valuable comment. We truly appreciate your suggestion to include performance indicators for different scales in the ablation experiments. In our current approach, we adhere to the Swin Transformer’s parameter configuration, where the input size is progressively reduced from 64×64 to 2×2 across several stages (64×64 → 16×16 → 8×8 → 4×4 → 2×2). This gradual downsampling allows for effective multi-scale feature extraction while preserving essential spatial details. Following this, we use transposed convolutions to restore the image back to its original size of 64×64.
We believe that this setup strikes a balance between computational efficiency and performance, particularly for the type of input data we are working with. However, we understand that investigating the impact of different scaling strategies could potentially yield further insights into the performance of our model. While we have not conducted such experiments in the current study, we fully recognize the importance of exploring how different scales could influence the model's performance in more diverse settings.
Your suggestion has highlighted an important direction for future research. In subsequent work, we plan to investigate the effects of varying scales on the fusion process to further improve the model’s robustness and generalization. Once again, we sincerely appreciate your thoughtful feedback, which will help us refine our research in the future.
Comment 5:In addition to the spatial error assessment, it is recommended that the authors give the spectral reconstruction errors.
Response: Thank you for your valuable comment. For the simulated datasets, we used the Spectral Angle Mapper (SAM) to evaluate the spectral quality, where lower SAM values indicate minimal spectral information loss. For the real ZY1E remote sensing dataset, as we do not have reference images, we provided spectral curve comparisons between the fused images from various methods and the original LR-HSI. These comparisons clearly illustrate the spectral reconstruction errors. Additionally, since the ZY1E dataset lacks reference images, we have also included the no-reference metric QNR to further evaluate the fusion results of different methods. We appreciate your insightful suggestion, which has helped improve the thoroughness of our evaluation.
Comment 6:It is recommended that parametric quantities and FLOPs for all methods be presented.
Response: Thanks for your comment. Regarding the comparison of model params and FLOPs, these are indeed important metrics for evaluating model performance and efficiency. We have added this comparison to the revised manuscript and included the corresponding analysis to highlight the advantages and characteristics of our model in terms of computational efficiency and performance compared to other relevant models.
Comment 7:The concluding section should contain the limitations of the proposed method.
Response: Thank you for your insightful comment. We acknowledge that upsampling low-resolution hyperspectral images for feature extraction and fusion may inevitably lead to some loss of information. In future work, we plan to explore alternative upsampling techniques, such as transposed convolution, to better align spatial resolutions and reduce information loss. We have incorporated a discussion of the limitations of our method, along with potential improvements, in the concluding section as per your suggestion. Your valuable feedback has been instrumental in enhancing the completeness of our manuscript, and we sincerely appreciate it.

Reviewer 3 Report
Comments and Suggestions for Authors
Summary
This work presents a model for the fusion of low-resolution hyperspectral images (LR-HSI) with high-resolution multispectral images (HR-MSI) to obtain high-resolution hyperspectral images. Previous methods based on convolutional neural networks (CNN) are mentioned, which are limited in their capacity and fail to fully account for the significant differences in scale and spatial resolution between the two types of images.
The proposed method, FCSwinU, incorporates a fast Fourier spectral convolution module to extract the feature representation of LR-HSI. Additionally, the self-attention window mechanism is used to extract global features from both HR-MSI and LR-HSI, exploring multi-scale features.
The model employs an encoder-decoder structure reminiscent of UNet to fully capture multi-scale spatial-spectral features. In the encoder, the Swin Transformer Feature Abstraction Module is introduced, while in the decoder, the Swin Transformer Feature Reconstruction Module is built to facilitate the recombination of features.
The current state of existing methods and the problem the proposed model aims to solve are described. The structure of the proposed model, the tests performed, and the results obtained are also detailed.
Experiments were conducted using five criteria to evaluate the model's performance, comparing it with eight other models. The tests were successfully applied to three datasets and one additional dataset.
I recommend accepting the paper, with possible minor changes at the editor's discretion. Based on the presented results, I consider the model to be a significant contribution to existing methods.
Abstract
The motivation for the work and its advantages over existing methods are described, as well as the most relevant structure of the model. The tests and results obtained on the three datasets are also mentioned.
Introduction
Various examples are provided of the difficulties in working with multispectral images, contrasted with the advantages of using hyperspectral images. Previous work is also mentioned, where attempts were made to improve the spatial resolution of hyperspectral images using information from multispectral images, including models based on traditional methods and CNN-based methods.
Methodology
The structure of the proposed model is detailed.
Suggestion: In Figure 3, subfigure (c) could be placed earlier, as it is described in the text before (a) and (b).
Observation: In line 360, the dimensions mentioned in the text do not match those indicated in the figures.
Experimentation and Analysis
The datasets used and the indices considered to evaluate the model's performance are clearly described. The importance of each of the model’s modules is evaluated, and their influence on the final result is analyzed. Additionally, a quantitative comparison of the performance with eight other models is carried out using different indices, along with a qualitative comparison of the results for the datasets and real data, highlighting the characteristics and differences between the evaluated models.
Conclusions
It would be helpful to explain the scope of the network regarding the dimensions in which it operates, whether there is any limitation concerning the dimensions of HR-MSI and LR-HSI images, and if there are any further tests they plan to implement to explore these limits.
Author Response
Dear reviewer,
Thank you very much for the opportunity to improve the quality of our manuscript, "FCSwinU: Fourier Convolutions and Swin Transformer UNet for Hyperspectral and Multispectral Image Fusion" (sensors-3145225). We sincerely appreciate your thorough and positive evaluation of our work, along with the constructive and insightful comments you have provided.
In response to your valuable feedback, we have carefully revised the manuscript and addressed all your suggestions. To assist in the review process, all revisions are highlighted in yellow in the updated manuscript. Additionally, we have provided a detailed point-by-point response to each of your comments, which you will find below.
We hope that the revised manuscript now meets the publication standards of Sensors, and we are truly grateful for your time and effort in helping us improve the quality of our work. Your feedback has been instrumental in enhancing the clarity, accuracy, and overall presentation of our research.
Comment 1:Methodology The structure of the proposed model is detailed.
Suggestion: In Figure 3, subfigure (c) could be placed earlier, as it is described in the text before (a) and (b).
Response: Thank you for your valuable suggestion. We appreciate your attention to the structure and flow of our manuscript. You are correct that subfigure (c) in Figure 3 is referenced earlier in the text compared to subfigures (a) and (b). To improve the clarity and coherence of the presentation, we will revise the figure order to align with the flow of the description in the text. This adjustment will make it easier for readers to follow the discussion.
Thank you again for your insightful feedback, which helps enhance the readability and organization of our manuscript.
Comment 2:Observation: In line 360, the dimensions mentioned in the text do not match those indicated in the figures.
Response: Thank you for your careful observation. We have reviewed the discrepancy between the dimensions mentioned in line 360 and those indicated in the figures. The necessary corrections have been made to ensure consistency between the text and the figures. We appreciate your attention to this detail, as it has helped us improve the accuracy and clarity of our manuscript.

Reviewer 4 Report
Comments and Suggestions for Authors
Below are comments on each paper section.
1. Introduction
The introduction begins by presenting the usefulness of finding a good solution to the image reconstruction problem by merging data at different resolutions. It clearly presents the difficulties encountered in this process. It presents and critically discusses in depth aspects of published works on the fundamentals. Finally, it clearly announces the contributions of the present work.
2. Related Work
This section presents in detail aspects of interest found in related works and announces the contribution of this work in relation to those, particularly as being the first that combines the Swin Transformer with a multi-scale structure and that applies it to the field of hyperspectral and multispectral image fusion.
3. Methodology
The section on methodology presents the mathematical basis well. It explains the structure of the proposed solution in good detail, with good figures. Nothing is missing.
4. Experiments and analyses
4.1. Datasets
The data used in the experiments were described in good detail.
4.2. Implementation and Indices
The parameters used in the experiments and the computational resources were appropriately described. The performance metrics to be used were also presented.
4.3. Ablation Experiments
Experiments were described, which proved that each module of the system was important for obtaining good results.
4.4. Results and Analysis
The experiments were well planned. They carefully demonstrated the superiority of the proposed method.
5. Conclusion
The conclusion reflected the results of the work well and makes interesting proposals for future work.
References
The list of references is comprehensive and up-to-date.
Therefore, it is concluded that the article is of high quality and deserves to be published.
Author Response
Dear reviewer,
Thank you very much for the opportunity to improve the quality of our manuscript, "FCSwinU: Fourier Convolutions and Swin Transformer UNet for Hyperspectral and Multispectral Image Fusion" (sensors-3145225). We sincerely appreciate your thorough and positive evaluation of our work, along with the constructive and insightful comments you have provided.
We are pleased to hear that you found the introduction, related work, and experimental design well-detailed and that the clarity of the methodology and analysis was up to your expectations.
Thank you again for your time and valuable feedback, which motivates us to further refine and improve our work in future endeavors. We are excited to share our findings with the community and hope that this work contributes meaningfully to the field.
Round 2
Reviewer 1 Report
Comments and Suggestions for Authors
(1) The introduction of the abstract provides a clear background on hyperspectral and multispectral image fusion, but it can be made more concise. For example, the first two sentences could be condensed into something like: “The fusion of low-resolution hyperspectral images (LR-HSI) with high-resolution multispectral images (HR-MSI) offers a cost-effective solution for obtaining high-resolution hyperspectral images (HR-HSI). Lack of qualitative evaluation in the abstract.
(2) To improve readability, consider simplifying some of the technical terms. For instance, you can say, “Our novel FCSwinU network leverages the SFFC module for spectral feature extraction and uses the Swin Transformer’s self-attention mechanism for multi-scale global feature fusion.
(3) The paragraph layout of the Introduction and Related Work sections needs to be improved.
(4)
Comments on the Quality of English Language
The quality of the English language is relatively good
Author Response
Dear Reviewer,
We extend our heartfelt gratitude for providing us with the opportunity to enhance the quality of our manuscript titled "FCSwinU: Fourier Convolutions and Swin Transformer Unet for Hyperspectral and Multispectral Image Fusion" (sensors-3145225). Your constructive and insightful comments have been invaluable to us.
We have diligently revised the manuscript in accordance with your feedback and suggestions. It is our aspiration that the revised version now aligns with the publication standards set by Sensors.
To streamline the review process, we have highlighted the revisions in the manuscript using yellow. Furthermore, we have included detailed point-by-point responses to address each of your comments effectively.
Thank you once again for your time and expertise in reviewing our work.
Comment 1:The introduction of the abstract provides a clear background on hyperspectral and multispectral image fusion, but it can be made more concise. For example, the first two sentences could be condensed into something like: “The fusion of low-resolution hyperspectral images (LR-HSI) with high-resolution multispectral images (HR-MSI) offers a cost-effective solution for obtaining high-resolution hyperspectral images (HR-HSI). Lack of qualitative evaluation in the abstract.
Response: Thank you very much for your thoughtful comments and valuable feedback. We appreciate your suggestions for the abstract. We have condensed the first two sentences to: 'The fusion of low-resolution hyperspectral images (LR-HSI) with high-resolution multispectral images (HR-MSI) offers a cost-effective solution for obtaining high-resolution hyperspectral images (HR-HSI).' Additionally, regarding the point about the lack of qualitative assessment in the abstract, we have now included a description of this qualitative assessment. Thank you once again for your constructive feedback.
Comment 2:To improve readability, consider simplifying some of the technical terms. For instance, you can say, “Our novel FCSwinU network leverages the SFFC module for spectral feature extraction and uses the Swin Transformer’s self-attention mechanism for multi-scale global feature fusion.
Response: Thank you for the suggestion. We appreciate your feedback on enhancing readability by simplifying technical terms. We have revised the text accordingly to make it more accessible to a wider audience without compromising the essence of the content.
Comment 3:The paragraph layout of the Introduction and Related Work sections needs to be improved.
Response: Thank you for the suggestion. We have restructured the paragraph layout of the Introduction and Related Work sections to enhance clarity and readability. We have ensured that the information flows logically and is presented in a more organized manner for better comprehension, using the present perfect tense. Thank you again for your continued feedback and guidance. Your input is greatly appreciated.
For detailed modifications, please refer to the highlighted sections in yellow in the revised manuscript.

Reviewer 2 Report
Comments and Suggestions for Authors
The author answered all my questions.
Author Response
Thank you for your feedback. I am immensely grateful for you taking the time from your busy schedule to review our manuscript, offering us the chance to improve the quality of our submission. Thank you sincerely for your review and careful guidance once again.